# A Systematic Review of Evidence-Based Alternative Models of Incarceration

**Anamalia Suʻesuʻe** [1,*], **Dylan Pilger** [2] **and Lorinda Riley** [3,4]

1 Department of Psychology, University of Hawaiʻi at Mānoa, Honolulu, HI 96822, USA
2 Independent Researcher, Nago 905-0005, Japan; dpilger@hawaii.edu
3 Office of Public Health Studies, University of Hawaiʻi at Mānoa, Honolulu, HI 96822, USA; lorindar@hawaii.edu
4 Kamakūokalani Center for Hawaiian Studies, University of Hawaiʻi at Mānoa, Honolulu, HI 96822, USA
* Correspondence: suesuea@hawaii.edu

**Abstract:** While much of the American justice system utilizes punitive models of sentencing and incarceration, restorative justice (RJ) approaches provide a holistic alternative to wrongdoing, viewing offenses in terms of relationships and paying particular attention to victim and community needs. These alternative RJ approaches have been shown to decrease recidivism and align with the values of those who have been most impacted by mass incarceration, including Indigenous populations. The purpose of this systematic review is to provide an overview of alternative models of incarceration utilizing RJ principles that could be adapted for a largely Indigenous population.

**Keywords:** restorative justice; Indigenous; incarceration

## 1. Introduction

America has the highest rate of incarcerated individuals in the world. This number may be reflective of the principles and values the current American criminal justice system is founded on. Much of the American justice system utilizes punitive models of sentencing and incarceration. Punitive models seek to identify the individual who committed the crime/wrongdoing and punish them with strict adherence to the letter of the law. However, within these models, little attention is paid to the victims of the offense and their needs. Moreover, the impact on the wider community is generally not considered, and plans for reintegration after punishment/incarceration are usually not fully addressed. Restorative justice approaches to incarceration may provide some benefits to society, yet there are few comprehensive studies that explore these alternative approaches to incarceration. This review provides an analysis of the scientific literature around alternative restorative approaches to incarceration.

### 1.1. Restorative Justice

Restorative justice (RJ) adopts a holistic approach to wrongdoing and views offenses in terms of relationships, paying particular attention to victims and their needs. RJ has been defined as "a process whereby parties with a stake in a specific offense collectively resolve how to deal with the aftermath of the offense and its implications for the future" (Marshall 1999). Marshall (1999) further identifies five primary objectives of RJ, including (1) attending to victims' and their support systems' needs (i.e., material, financial, emotional and social), (2) reintegrating offenders into the community, (3) enabling offender accountability and responsibility for their actions, (4) recreating a "working community" to support

offenders and victims, and (5) avoiding escalation in the justice system (i.e., costs and delays). Meanwhile, the current punitive justice system attempts to resolve a dispute at arm's lengths, thereby undervaluing the community (Riley 2020). In a conventional retributive justice system, the two parties involved are "positioned as adversaries, discouraged from communicating directly with each other, and expected to remain passive whilst all the key decisions are made by professionals." (Johnstone 2003, p. 2). Because the stakes are high in a conventional approach and because there is little inclusion of the context of a criminal act, including the personal background of the offender, there is little incentive for the offender to take responsibility for their actions. In fact, the system incentivizes providing as little information as possible to cast a "reasonable doubt" of their guilt.

Rather than focusing on punishment, which RJ advocates hold does not always work alone, restorative models seek to instill remorse on the part of the offender and make the victim whole (Braithwaite 2014; Zehr 2002). In making the victim whole, reparations may be provided to not just the victim, but also the larger community. The act of repairing harm has an added benefit of aiding offenders to reintegrate into the community by providing an opportunity to regain the community's respect (Dzur 2003, p. 6). The core goals of RJ include healing the victim, repairing relationships, holding offenders accountable, increasing community stakeholder involvement, and reintegrating offenders as part of society (Zehr 2005). Moreover, because the harmful act is construed as being perpetrated by one person against another person, the victim is fully engaged in each step of the process (Braithwaite 1999, pp. 21–23). In contrast, the current punitive model of justice, by focusing solely on the criminal act rather than on the lives of the offenders, their other circumstances often forgotten, leaves the justice system open for high rates of recidivism and overcrowded prison facilities.

### 1.2. Incarceration and Indigenous Overrepresentation

Globally, Indigenous communities are overrepresented in the carceral system. The United Nations has long acknowledged the practice of over incarceration of Indigenous populations (Economic and Social Council 2024). Among the larger settler colonial nations, such as Canada, nearly 40% of incarcerated individuals identify as Indigenous, despite making up only 5% of the population (Malakieh 2018; Statistics Canada 2019). Similarly, Australian Aboriginal people were 28% more likely to be arrested than non-Aboriginals in 1996 and in 2024 Aboriginal youth were still 28% more likely to be detained overnight than non-Aboriginal youth (Australian Institute of Health and Welfare 2024; Office of the Aboriginal and Torres Strait Islander Social Justice Commissioner 1996).

The United States has a long history of disenfranchising minority groups, none more evident than outcomes in the legal and justice systems. Many of those currently incarcerated are from poor, Black, and Indigenous communities. In particular, Indigenous communities across the US have endured generations of overrepresentation in the American prison system (Cunneen and Tauri 2017). In 2021, the national incarceration rate in state and federal prisons for all US residents was 350 per 100,000 (Carson 2021). However, for American Indian and Alaska Natives, this rate was more than doubled at 763 per 100,000 (Carson 2021).

The impacts of punitive US models of incarceration are felt thousands of miles away from the American continent in Hawai'i, where Native Hawaiians are overrepresented in both juvenile and adult correctional facilities. The Hawai'i Department of Public Safety's 2022 Annual Report noted that in October 2022, there were over four thousand (4287) offenders in Hawai'i's correctional system (State of Hawaii Department of Public Safety 2022). Of those, 1571 were Native Hawaiian, making up approximately 37% of the population, despite making up approximately 20% of the population statewide (U.S. Census Bureau 2022).

Continuing to ignore the wider context of offenses, including their far-reaching impacts, may compound the problems of the current overpopulated incarceration system.

International Integration of Indigenous RJ

Though RJ alternatives in Western justice systems are relatively new, these practices are longstanding in Indigenous communities, making RJ an apt approach for these populations. Restorative approaches that refer eligible offenders to community-based treatment or provide RJ services during their sentences have been shown to have positive effects. In addition to the positive outcomes that RJ alternatives have provided for offenders and communities, these approaches also align with the values of those who have been most impacted by mass incarceration, including Indigenous populations. Before Western contact, many Indigenous communities addressed wrongdoing or harm within the context of their respective tribe or familial units. For example, for Native Hawaiians, ʻohana (family) is a central value, and maintaining ʻohana has been critical for survival, especially while navigating the American justice system (Friesema 2013). Hoʻoponopono, a traditional family-based dispute resolution method, has been integrated into Western systems to repair harm within the ʻohana and support formerly incarcerated individuals (Hawaii State Judiciary 2018). Family and community involvement is essential in harm repair for Native Hawaiians and Indigenous populations, coinciding with RJ principles of stakeholder involvement (Maxwell and Hayes 2006).

Australia, like other settler colonial nations, has recognized the disparate impact of the justice system among the Aboriginal population (Harris 2006; Tomaino 2004). One modern approach has been to institute specialized sentencing courts, with the first such court being established in Nunga, South Australia, in 1999. These courts provide a more culturally appropriate sentencing practice while simultaneously ensuring the engagement of Indigenous communities in the justice environment. While RJ-oriented actions, such as family conferences, cannot be seen as a panacea (Little et al. 2018), there have been a variety of innovations integrating traditional approaches into the Western justice system (Marchetti and Daly 2007). One study found that Murri Courts, which incorporate elders in a more informal setting that integrates significant background information on the offenders, have seen lower failure to appear rates and may have increased community support of offenders (Morgan and Louis 2010).

Similarly, in New Zealand, whanau (extended family) and respective tribes are central to wellbeing for Indigenous Maori (Marques et al. 2021). Traditionally, when an individual offended, it was attributed to a lack of balance in the offender's social unit, which led to a collective responsibility to redress those actions. As such, programs within the New Zealand criminal justice system have adopted Indigenous Maori principles that often coincide with RJ principles in their work collectively with Maori offenders (Toki 2018; Tauri and Morris 1997). For example, the New Life Akoranga Program, which focuses on general criminality among Maori offenders, incorporates features such as Maori language and values, as well as including Maori chiefs in the process, reflecting RJ principles of community involvement (Gutierrez et al. 2018).

Finally, while the Canadian justice system is unique, they have integrated RJ approaches on several levels. First, through the courts, advocates are able to submit Glaude reports, which allow for the introduction of the offender's background for the purposes of sentencing. Second, sentencing circles consisting of Indigenous community members and other stakeholders collaboratively determine a sentence for an offender and make a recommendation to the judge, who may or may not accept that recommendation (Department of Justice Canada 2016). Third is the integration of Indigenous programs within the standard carceral system, which when evaluated suggested the building of cultural pride,

religious connection, and digitalization among the Indigenous incarcerated population (Tetrault 2022). Finally, the development of Healing Lodges, which incorporate cultural and healing practices into programming to support the healing and reintegration of offenders (Nielsen 2003). These practices highlight the global movement toward integrating RJ principles into correction to promote healing.

*1.3. Alternative Incarceration Models*

While an abundance of international efforts exist to integrate RJ principles into the carceral environment, this practice has yet to take hold in the majority of US jurisdictions. These models aim to change what prison has typically looked like, whether it be in terms of physical or structural design or routine, in order to make lasting impacts. Some approaches have been implemented in traditional prison facilities, while others are based in community settings. For example, among those alternative models in prisons, therapeutic communities adopt a "community as method" approach and usually require entirely separate wings of facilities, requiring individuals to adhere to strict daily routines while also allowing them freedom to roam across different areas of the wing (Weinrath et al. 2021). Other community-based approaches include RJ conferencing options for misdemeanor or felony sex crimes that focus on offender accountability and agreement development to repair harm (Koss 2014).

The increased use of nature or biophilic designs in prison have also been used to change the look of facilities while supporting positive inmate behavior (Söderlund and Newman 2017). While these approaches have gained some momentum in the international community, the United States' heavily punitive structure makes outdoor engagement logistically difficult to implement (Reddon and Durante 2019; Moran and Turner 2019). In some cases, alternative approaches to incarceration with RJ principles have quantitatively been shown to decrease recidivism (Richner et al. 2023). While recidivism is not the only measure of success, lowering recidivism theoretically decreases overall carceral load (Rosenfeld et al. 2022). For example, among probationers who participated in a brief RJ intervention (RJI) and those who received treatment as usual, the RJI participants had lower recidivism rates in the long term and less reoffending compared to the control group, thereby reducing overcrowding (Kennedy et al. 2019). The purpose of this paper is to provide an overview of alternative restorative models of incarceration that could be adapted for a largely Indigenous population.

## 2. Methods

To identify existing alternative incarceration models, our research team conducted a systematic review of the literature. Our research team consisted of a Principal Investigator (PI) and two graduate research assistants. We searched three databases, PubMed, PsycNet, and Google Scholar (See Table 1 for search terms), resulting in a total of 1904 records after duplicates were removed. We used Cadima, an online tool for systematic reviews. Two research team members independently reviewed titles and abstracts for relevancy. The review of records began with a consistency check of the criteria on 10% of the records (n = 190). The results of the consistency check were a Kappa value of 0.7216796875 or a "good" strength of agreement. The next two rounds of the review included the application of criteria, first to title/abstracts, followed by full-text records. A portion of the records in each round underwent parallel assessment or were assessed by both reviewers, one round with 40% of records in the title/abstract (n = 761) and one with 20% (n = 171) in the full text. Conflicts between reviewers were discussed and resolved through consensus.

Our inclusion criteria for eligible papers included (1) a focus on Indigenous, restorative, or therapeutic justice, (2) sufficient information on an alternative incarceration model

to be recreated, (3) written in English, and (4) published between 2012 and 2022. In our review, we included both peer-reviewed and "gray" literature (i.e., governmental reports, research outcomes, investigative journalism, etc.). Studies that did not meet basic scientific rigor (e.g., studies that were not replicable due to a lack of information or where the evaluation criteria was susceptible to bias, especially in participant selection) were not included. After several rounds of review, a total of 120 articles remained for data extraction (See Figure 1 for the flow diagram).

Data were extracted from 120 articles in order to screen for the final inclusion criteria, requiring sufficient information on an alternative incarceration model for potential replication. To be considered sufficient, data were needed in the following areas: intervention components, restorative justice components (i.e., as outlined by the aforementioned definitions of RJ), physical design elements, partnering agencies, and information on findings. A total of 41 articles met this criteria and were then combined into eight categories of analysis (see Table S1 for a table of studies and Table S2 for intervention details on FigShare—to be added). We followed the PRISMA 2020 guidelines but opted not to register the study.

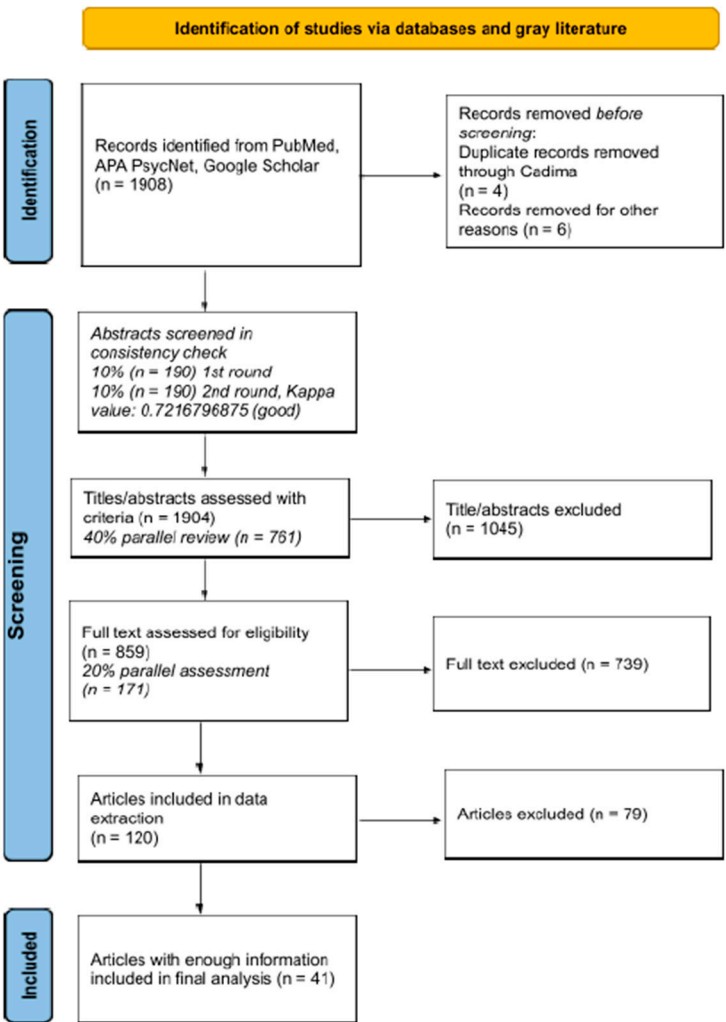

**Figure 1.** PRISMA flow diagram for RJ/alternative incarceration model systematic review.

**Table 1.** Databases and search terms for systematic review of RJ/alternative incarceration models.

| Database | Search Terms | Year Range | Results |
|---|---|---|---|
| PubMed | "Indigenous" and "justice" in abstract | 2012–2022 | (n = 306) |
| | "restorative" and "justice" in abstract | 2012–2022 | (n = 146) |
| | "therapeutic" and "justice" in abstract | 2012–2022 | (n = 301) |
| APA PsycNet | "Indigenous" and "justice" in abstract, peer review journal | 2012–2022 | (n = 260) |
| | "restorative" and "justice" in abstract, peer review journal | 2012–2022 | (n = 381) |
| | "therapeutic" and "justice" in abstract, peer review journal | 2012–2022 | (n = 371) |
| Google Scholar | "Indigenous and "court" in title | 2012–2022 | (n = 119) |
| | "Indigenous and "restorative justice" in title | 2012–2022 | (n = 24) |

## 3. Findings

Upon analyzing the 41 articles for common themes, eight categories emerged. The eight categories are described below and include therapeutic communities, nature-based programs, educational interventions, victim–offender mediations, Indigenous-based models, Circles of Accountability and Support, victim impact panels, and other. Under the definition of RJ described above by Marshall (1999), each of these eight categories are restorative.

### 3.1. Therapeutic Communities

Five articles from our search discussed therapeutic communities within correctional facilities. The category of therapeutic communities (TCs) is defined as those practices that utilize a "community as method" approach and follow principles such as choice, responsibilities, and routine. In our review, the articles included in the TC category included RJ components of offender accountability, repairing harm, and restoring positive relationships with the community. Weinrath et al. (2021) described TCs as a unit-level way to "promote positive prison environments", further adding that TCs "are separated physically from other areas and provide structure and an interactive milieu for offenders to support each other, as well as encouraging interaction with staff". The structure and routines of TCs are designed to help participating offenders adopt more positive, prosocial behaviors.

The TCs from our search were often designated for a specific group of offenders. For example, Wilson and Brookes (2020) and Bennett and Shuker (2017) described TCs for serious violent or potentially violent offenders, the Barlinnie Special Unit (BSU) and HM Prison Grendon. In these TCs, regular group meetings within the unit were used to designate job assignments and discuss and resolve issues, as well as electing "chairs" for meetings. Regular group therapy sessions were also part of these TC routines. Though the BSU was discontinued, HM Prison Grendon continues to serve as the UK's only TC prison, with each of its six wings acting as a separate TC for approximately 230 residents.

Other TCs were designed for offenders convicted of sex crimes against minors. Frost (2017) outlined the Kia Marama in Aotearoa, a 60-bed TC for adult men convicted of sexual offenses against minors and actively seeking change towards an abuse-free life. Kia Marama has regular group therapy sessions, which are held in a separate building next to the prison unit for two and a half hours a day, three times a week, for approximately thirty weeks. A TC for offenders with substance use disorders in Canada was described by Weinrath et al. (2021). Similar to other TCs, this TC had routines, leadership roles and responsibilities, and regular group meetings to track progress.

Lastly, Bainbridge (2017) described a psychologically informed planned environment (PIPE) in a women's prison, which was similar to TCs in its focus on staff and peer relationships, activities, and community. The PIPE aims to achieve better outcomes for participants by "increasing pro-social skills, creating a calm and safe environment and developing a workforce that has appropriate skills and confidence" (Bainbridge 2017, p. 173). Focus groups with women who had been in PIPEs found that the relaxed environment, supportive staff, ordinary activities, and belongingness with peers were all important factors in their experience.

### 3.2. Nature-Based Programs

Four articles from our search focused on nature-based programs in prisons. The category of nature-based programs is defined as articles describing programs that incorporate some form of animal- or plant-based work or exposure for offenders to facilitate healing, promote health, or develop skills for employability. In our review, the articles included in the nature-based category included RJ components of community reintegration and promoting positive relationships with the community. For example, the articles included a program on therapeutic gardens and outworking opportunities for prisoners in the UK (Baybutt et al. 2019). Baybutt et al. (2019) described the "Greener on the Outside for Prisons" (GOOP) program, which was composed of both nature-based activities in the community for prisoners on temporary release as well as in-prison nature-based activities. The in-prison work focused on the development, maintenance, and design of horticultural spaces and therapeutic gardens for both prisoners and staff, growing food and plants, and participating in accredited job and skills training. Baybutt and colleagues found that benefits of the GOOP program fell into three overarching themes, including (1) health and wellbeing (i.e., improved mental health, physical activity, eating habits), (2) skills development, employability, and work preparedness (i.e., communication, teamwork, mindfulness, etc.) and (3) relationships (i.e., positive prisoner–prisoner and prisoner–staff relationships). Other articles in the nature-based programs category included reviews of animal-work and gardening programs with offenders (Moeller et al. 2018; Payne et al. 2022) and changing the design of prisons to incorporate more greenery and nature (Söderlund and Newman 2017).

### 3.3. Educational Interventions

Five articles from our search described educational interventions implemented in correctional facilities. In our search, we categorized educational interventions as those consisting of a structured curriculum taught weekly over the course of several months, though one program consisted of one brief 8 h educational session (Kennedy et al. 2019). Across the programs, the content mainly focused on developing awareness and accountability for the impacts that offenders' crimes had on victims, communities, and others like the offenders' families. In our review, the articles included in the educational interventions category included RJ components of offender accountability, including increasing awareness of the impact of crime, repairing harm, and stakeholder involvement. Activities to practice accountability included reading victim impact statements, sessions with victim guest speakers, and writing apology letters to victims.

Some educational interventions took a more tailored approach and focused on specific subpopulations of offenders or particular approaches, such as offenders who use substances (Hechanova et al. 2020) or faith-based educational programs (Armour and Sliva 2018). For example, in their educational intervention for offenders who use methamphetamine, Hechanova et al. (2020) described different modules in the intensive 22+ week program, including material focused on drug recovery skills (i.e., coping with cravings,

managing external triggers). The modules also incorporated restorative principles such as rebuilding relationships and family modules, where family members discuss, plan, and write a contract to help the offender and family through reentry and recovery.

Other educational interventions focused more generally on repairing harm and restoring relationships after crimes (Folk et al. 2016; Sedelmaier and Gaboury 2015). For example, Folk et al. (2016) described the eight week Impact of Crime program implemented in a county jail setting, which aimed to encourage offenders to understand ways that crimes impact victims, the community, and others. Rather than tailoring content to one type of crime, the sessions introduced offenders to the broad idea of RJ and covered the impacts of various types of crimes.

*3.4. Victim–Offender Mediations*

The largest category from our search focused on victim–offender mediations or restorative justice conferences, with nine articles describing these processes. Though there was some variability across the articles, this category is defined by certain shared tenets, including preparation and facilitation by a trained mediator, the voluntary agreement/participation of the victim, the involvement of other stakeholders, including supports for the victim, discussions focused on the impact of crime and offender accountability, and a plan or agreement developed for the offender to complete to repair harm (sometimes overseen by a government entity—police, prosecutor, probation, etc.). In our review, the articles included in the victim–offender mediations category included RJ components of offender accountability, including increasing awareness of the impact of crime, centering victims' needs, repairing harm, stakeholder involvement, restoring relationships, and community reintegration.

One article provided a review of the efficacy of RJ conferences (RJCs) in reducing recidivism (Sherman et al. 2015). In their review of ten studies, Sherman et al. (2015) found that RJCs were likely to reduce recidivism among participating offenders and that consent from all parties, as well as the degree of emotional discussion, may also contribute to participation in RJCs. There were several articles that focused on mediations or conferences designed for specific types of crime or a specific severity of crime. For example, Mills et al. (2013) described the Circles of Peace program, which focused on repairing harm for offenses involving domestic or intimate partner violence, whereas Koss (2014) outlined the RESTORE program, which focused on offender accountability and agreement development to repair harm for misdemeanor or felony sex crimes. Beck et al. (2015) described how RJCs may be an optimal justice option for crimes against older adults, providing older adult victims with a sense of control and power in their lives. Both Stewart et al. (2018) and Walters (2015) described how RJCs may be used for serious crimes, including homicide, finding that participating in RJ options was related to statistically significant improved results for offenders while under conditional release (Stewart et al. 2018) and improved emotional wellbeing reported by the family members of victims (Walters 2015).

Lastly, three articles described programs from Hawai'i or the influence of Hawai'i programs in their RJ models. Lehmann et al. (2012) described solution-focused brief therapy (SFBT) in the criminal justice system, which uses a forward- and goal-oriented, strengths-based approach, with significant contributions to SFBT made by Lorenn Walker and E Makua Ana Youth Circles in Hawai'i. In these youth circles, foster children were able to develop goals and plans as they transitioned out of care. Another RJ example from Hawai'i, the Huikahi Restorative Circle program, was described by Hass and Saxon (2012). The Huikahi Restorative Circle program was designed for incarcerated offenders and their family to discuss the impacts of crimes and develop plans for reentry and harm repair. Finally, Pennell et al. (2021) described family group conferencing or 'Ohana Conferences in

Hawai'i's EPIC 'Ohana, Inc. (EPIC 'Ohana, Inc. n.d.) 'Ohana Conferences bring together families who have been referred to Child Welfare Services, along with their community supports, to develop plans for their children's safety (Adams and Chandler 2002).

*3.5. Indigenous-Based Models*

Six articles discussed Indigenous-based justice models or programs and their implementation in the justice system. We use the term Indigenous here to refer to the cultural systems and practices of the earliest known peoples in a place and their descendants, particularly for those places that have been now settled or occupied by colonial powers. Several articles described Indigenous restorative processes at the court level. For example, Yuzicapi (2013) outlined sentencing circles and proposed a restorative justice center where these processes might take place, while Horn (2016) and Daly and Marchetti (2012) provided overview descriptions of Indigenous justice and their overlap and divergence from restorative justice. In our review, the articles included in the Indigenous-based models category included RJ components of offender accountability, repairing harm, stakeholder involvement, restoring relationships, and community reintegration.

Two articles described programs for justice-involved Indigenous offenders upon release. Lau et al. (2012) described the Gathering Place Health Service (GPHS), a pilot program in Australia providing traditional healing services for justice-involved Indigenous Australians, many with coexisting substance use and mental disorders. Traditional healing services provided at GPHS included weekly healing circles led by Indigenous elders to help participants connect with their Indigenous spirituality and culture. Separate healing circles were held for men and women and consisted of "a welcome or acknowledgement to country, a traditional smoking ceremony to cleanse the body of negative energy and evil spirits and discussion about family names, country/heritage and spirit totem as well as traditional Indigenous music" (Lau et al. 2012). Though Lau et al. (2012) note that no formal evaluation has been presented for GPHS, success stories of past participants offer a promising outlook for this program.

Gutierrez et al. (2018) briefly outlined the Te Whanau Awhina Program, which provides restorative justice opportunities at Hoani Waititi Marae, with restorative processes for serious crimes held in a Wharenui (a traditional Maori meeting house). In this program, offenders gather with a panel of marae members as well as with their family and other supporters to develop a plan for harm repair and restoration. Other research on this particular program found that offenders who participated in Te Whanau Awhina reported accepting decisions made by the panel and found being on the marae more meaningful, feeling a closer connection to their ancestors (Maxwell and Hayes 2006).

One article described Indigenous healing programs in a correctional facility. Perdacher et al. (2019) outlined monthly sweat lodges conducted by the Native Sisterhood group at a women's prison in Canada and the impact on Indigenous participants. Other studies about these sweat lodges described them as follows:

> "…complete darkness, cramped quarters, and a heat so intense that it burned the skin. During the ceremony, water was poured on the Grandfathers (hot rocks) and the steam that rose ran through the nose and deep into the lungs. Every emotion was felt in the Lodge–from gut-wrenching sobs, to songs of courage, and lullabies of peace. The physical pain in the Sweat facilitated the release of emotional pain and ultimately relief". (Yuen and Pedlar 2009)

Incarcerated participants of the Native Sisterhood sweat lodges reported support for emotional healing as well as a preference to address wellbeing issues with Indigenous practitioners and elders using Indigenous practices (vs. non-Indigenous approaches)

(Perdacher et al. 2019). With this, the sweat lodge ceremonies offered incarcerated women an opportunity to engage in both cultural and emotional healing.

### 3.6. Circles of Accountability and Support

Three articles from our search focused on a specific program, Circles of Accountability and Support (COSA or "Circles"). This category, therefore, focuses on articles about COSA, a community-based program designed for high-risk sex offenders who may voluntarily participate in a Circle after completing their sentences. Offenders are considered the "core members" of the Circle, which is composed of four to six community volunteers. In our review, the articles included in the COSA category included RJ components of offender accountability, repairing harm, and stakeholder involvement.

The Circle meets weekly to oversee core member transition back to society and offers assistance as needed. The core member is accountable to the Circle, which can also include law enforcement or legal entities such as parole officers who may use their discretion to return core members to prison if they violate the terms of the circle. Azoulay et al. (2019) provided a review of international COSA implementations across countries including the United States, New Zealand, Scotland, and the United Kingdom. This review found that there was a reduction in recidivism and costs in pilot circles in Canada, the United States, and the United Kingdom.

Other reviews of COSA implementations have found promising results of the program. Clarke et al. (2017) also conducted a review of COSA outcomes internationally and found that there were few significant differences between core members and controls, but when differences were present, core members did better. Additionally, their review reported there was currently no evidence that Circles have adverse effects. Elliott and Zajac (2015) examined COSA in the United States and summarized that COSA must balance a "flexible, responsive nature" along with needs to be evaluated in order to garner support from the criminal justice system and policy makers. COSA might face challenges with evaluation, including differences in outcome selection and implementation or core member selection issues.

### 3.7. Victim Impact Panels

Three articles described victim or surrogate impact panels. The category of victim impact panels is defined by panels that bring offenders and a panel of victims of a certain crime type together for a discussion on the impacts of the crime. For example, the panels included in the three articles in our search were for victims and offenders of intimate partner violence or domestic violence. The panel of victims present their stories and experiences to the group of offenders, most of which have been mandated to attend the session. Panels are usually held once a month in a large community space, including a community center or hall, are led by a trained facilitator, and last between one to two hours. In our review, the articles included in the victim impact panels category included RJ components of offender accountability, including increasing awareness of the impact of crime, repairing harm, centering victims' needs/voice, and stakeholder involvement.

Zosky (2018) described quarterly panels that were coordinated by a county probation office and facilitated by the probation staff. In these panels, the offender audience was not allowed to engage with the victims during the panel presentation, though after the speakers shared their stories, the facilitators worked with the offenders in small groups to process the session. Offenders, also referred to as Justice-Involved Individuals (JIIs), were also required to complete a Batterer Intervention Program (BIP). Similarly, in their studies on surrogate impact panels, Kerrigan and Mankowski (2021a, 2021b) also stated that JIIs on the respective panels were mandated to attend the session and had to complete a BIP;

however, one of these panels allowed for a question-and-answer session between the panel and audience. Studies have found that panels have a positive effect on JIIs, including an emotional impact (i.e., feeling humbled or sobered) and influencing their intent to change (Kerrigan and Mankowski 2021a), as well as an increased awareness of the impact of the crime on victims (Zosky 2018).

*3.8. Other (Juvenile, Therapeutic Jurisprudence)*

Six articles discussed models or programs with RJ principles within court or juvenile populations. These articles fell into the category of "other" due to the key features of their practice, such as the age of the target population or using a court-level intervention, and they differed greatly from the previous seven categories; yet the practice still included information on interventions and RJ principles that may prove helpful when considering adaptations. In our review, the articles included in the other category included RJ components of offender accountability, repairing harm, stakeholder involvement, restoring relationships, and community reintegration. Three articles focused on justice-involved youth and community reintegration. One study used photo voice to explore a community juvenile diversion program (McMahon and Pederson 2020). Another study compared outcomes for justice-involved youth who participated in restorative community service to those who did standard community service, finding that those who participated in the restorative program were associated with more positive attitudes and peer relationships as well as less negative behaviors at school (Church et al. 2021). Finally, Holler (2019) described an RJ community arts project where youth probationers worked together with local organizations to paint a mural in their community.

While the three aforementioned articles were community-based, two articles discussed programs in juvenile correctional facilities. Carl et al. (2020) described social-therapeutic units (STUs) in Germany for juvenile offenders of serious crimes, while Elliot et al. (2018) detailed the Athletes Targeting Healthy Exercise and Nutrition Alternatives (ATHENA) intervention implemented in a female juvenile correctional facility that aimed to promote skills including emotional and physical self-efficacy and autonomy. Lastly, Bartels (2019) described therapeutic jurisprudence in Hawai'i's probation processes, including the expectations and sanctions of probationers.

# 4. Discussion

This systematic review examined the existing literature on alternative incarceration models with restorative justice principles. We identified 41 articles that met our criteria and described alternative incarceration programs or models (see Table S1—Table of studies included in the systematic review), which were categorized into eight categories: therapeutic communities, nature-based programs, educational interventions, victim–offender mediations, Indigenous-based models, Circles of Accountability and Support, victim impact panels, and other (i.e., therapeutic jurisprudence). Our results indicate that a variety of models exist that could be adapted to promote healing. In our review process, we included articles that included sufficient information on an alternative incarceration model to be recreated as well as those that included intervention components, physical design elements, partnering agencies, and information on findings (see Table S2—Details of RJ interventions/models included in review). Adapting models identified in the literature would require making strategic modifications to the model in alignment with the culture of the target population. This is in contrast to a "ground up" or culturally grounded approach that starts from content that is familiar and meaningful to program participants (Okamoto et al. 2014).

The overrepresentation of Indigenous people in the criminal justice system is a significant problem. The purpose of this study was, in part, to identify alternative models to incarceration that could be adapted to Indigenous communities. Several alternative models identified in this study that aligned with the existing values of an Indigenous community could easily be adapted. For example, Native Hawaiians value mālama ʻāina (caring for the land), and models within the nature-based programs category may be particularly well suited for Native Hawaiians. The therapeutic gardens described by Baybutt et al. (2019) can be adapted using local plants, including kalo (taro) or uala (sweet potato). To end youth incarceration in Hawaiʻi, the Kawailoa Youth and Family Wellness Center, once solely known as the Hawaiʻi Youth Correctional Facility, has transformed its campus to include a variety of restorative programs, many of which incorporate Native Hawaiian values of mālama ʻāina, such as Kupa ʻĀina Farms run by the Partners in Development Foundation (Opportunity Youth Action Hawaiʻi 2022). Programs such as these should be evaluated and expanded if successful.

Our study has several important limitations. First, we looked for programs that were evidence-based, which may exclude some models that have not been published in peer-reviewed journals. We also limited our review to studies in English, which may have left out some innovative models across the globe, including in Asia. Finally, because we wanted to focus on incarceration, we did not include purely court-based models. Making changes to the court or judicial system often requires legislative changes. Though court or judicial level changes would be ideal to support restorative justice incarceration models, this political process poses numerous pragmatic challenges. The models in this study, on the other hand, could be adopted by state correctional systems.

The results from our review extend beyond purely programmatic considerations to a space redesign process, which seeks to reduce recidivism by increasing healing within the correctional system. In fact, many alternative models to incarceration require a special space that facilitates the healing process or engagement with key stakeholders. Current designs for correction facilities are quite militarized/panopticon-esque and do not allow for these spaces. While overhauling the judicial system may not be feasible, creating space for healing is something that can be integrated into the corrections environment.

## 5. Conclusions

Existing alternative models to incarceration, such as RJ models, have shown promising results and offer a pathway for wider community healing. Populations that have been particularly marginalized by the current criminal justice system, like Indigenous communities, may benefit from these RJ alternatives as many RJ principles align with Indigenous values and practices. Considering the diverse and distinct characteristics of Indigenous populations in the United States, adaptations to existing RJ programs may be needed for meaningful change to be made. This rigorous review of the scientific and gray literature for potential RJ alternatives may serve as an initial step in centering healing in our justice system and restoring communities.

**Supplementary Materials:** The following supporting information can be downloaded at https://www.mdpi.com/article/10.3390/laws14020011/s1: Table S1 for table of studies and Table S2 for intervention details.

**Funding:** This research was funded by the Hawaiʻi Department of Corrections and Rehabilitation.

**Institutional Review Board Statement:** Not applicable.

**Informed Consent Statement:** Not applicable.

**Data Availability Statement:** The original data of included articles presented in the study are openly available at zenodo.org at https://10.5281/zenodo.14903424 accessed on 13 February 2025.

**Conflicts of Interest:** The authors declare no conflict of interest.

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
