# Peer review of "A Systematic Review of Evidence-Based Alternative Models of Incarceration"

_laws_

Round 1
Reviewer 1 Report
Comments and Suggestions for Authors
Introduction and Literature Review
A few minor points below:
- RJ advocates hold that punishment does not work (line 39). This claim should be expanded or at the very least cited.
- “…community stakeholder involvement, and reintegrating offenders back into society” (parallel construction with “ing”)
- Lines 83-88: Citations needed
- “alternatives of incarceration” (line 103) or “alternatives to..” (line 413) or “alternatives for” (lines 89, 90). These all have slightly different meanings and it would be more clear if the authors picked one and stayed consistent with it.
- I was unable to read Figure 1 because it was blurry
Methods
- “Studies that did not meet basic scientific rigor were not included” (lines 125-126). How was “scientific rigor” operationalized?
- “Data were” (line 128; 130)
Findings
It is difficult to draw conclusions from the findings other than the very broad conclusion that there are some alternatives to mainstream incarceration practices.
Author Response
Reviewer 1 Comments
We appreciated Reviewer 1’s comments. Below we address each of the comments.
- RJ advocates hold that punishment does not work (line 39). This claim should be expanded or at the very least cited.
Response: We have provided two citations and have also revised the language to more accurately described our intended meaning.
- “…community stakeholder involvement, and reintegrating offenders back into society” (parallel construction with “ing”)
Response: We revised the sentence as recommended (i.e., parallel construction with “ing”).
- Lines 83-88: Citations needed
Response: We have added two citations on family as a critical for Indigenous populations in harm repair and family as stakeholder involvement.
- “alternatives of incarceration” (line 103) or “alternatives to..” (line 413) or “alternatives for” (lines 89, 90). These all have slightly different meanings and it would be more clear if the authors picked one and stayed consistent with it.
Response: Thank you for bringing this to our attention. After review and discussion, we have changed these to "alternatives to" and ensured consistent use.
- I was unable to read Figure 1 because it was blurry
Response: We have revisited Figure 1 and provided a cleaner version in the manuscript.
- “Studies that did not meet basic scientific rigor were not included” (lines 125-126). How was “scientific rigor” operationalized?
Response: We operationalized scientific rigor as articles that clearly outlined the methods used in respective studies or reports.
- “Data were” (line 128; 130)
Response: We made this grammatically correct.
- It is difficult to draw conclusions from the findings other than the very broad conclusion that there are some alternatives to mainstream incarceration practices.
Response: The authors believe that our findings represent a valuable contribution to the broader field of law. While those who work within the RJ field may be familiar with these categories, the broader discipline has yet to recognize these as staples within the justice system. Moreover, because this review incorporated a systematic and rigorous approach that resulted in the exclusion of many studies, this review represents a compendium of evidence based programs. Additionally, while we were able to find many reviews discussing RJ and incarceration alternative models we were unable to find any that provide as comprehensive as this manuscript. The following literature review by the Department Of Justive does not include all the categories of our systematic review, highlighting the novelty of our manuscript which brings together numerous models in one place. https://ojjdp.ojp.gov/sites/g/files/xyckuh176/files/media/document/restorative_justice.pdf
And https://ecommons.luc.edu/cgi/viewcontent.cgi?referer=&httpsredir=1&article=1008&context=socialwork_facpubs
Reviewer 2 Report
Comments and Suggestions for Authors
Thank you for the opportunity to review this paper. I found it to be quite interesting. I felt that while the systematic review and its findings were very informative, more is needed in the literature review section. I felt that the authors have ignored a large amount of international literature on the subject of restorative justice. They could / should consider the some of the different models of RJ as well as the different setting eg police led, prison based, diversion for youth offender. I also felt that given that a significant focus of the paper was on indigenous peoples, the authors should make some reference to indigenous criminology and the work of Cunneen and Tauri in this area. This would also set the paper up for discussing the use of RJ in these indigenous communities. Again this would need to have a more international focus with reference to indigenous people in the US, Canada, Australia and New Zealand. These revisions / additions are relatively minor so I do hope that the authors are not disheartened by them.
Author Response
Reviewer 2 Comments
We appreciate the thoughtful comments of Reviewer 2.
- Thank you for the opportunity to review this paper. I found it to be quite interesting.
Response: Thank you for your interest in this topic and our findings.
- I felt that while the systematic review and its findings were very informative, more is needed in the literature review section. I felt that the authors have ignored a large amount of international literature on the subject of restorative justice. They could / should consider the some of the different models of RJ as well as the different setting eg police led, prison based, diversion for youth offender.
Response: Thank you for this comment. We agree that including international literature on restorative justice is important to create a comprehensive overview of the topic. In our initial search of the literature we were intentional in including studies from international settings that were available in English. These international studies were described throughout our findings and were important as part of our recommendations for alternative models for Native Hawaiians such as the therapeutic gardens in the UK as described by Baybutt and colleagues (2019).
- I also felt that given that a significant focus of the paper was on indigenous peoples, the authors should make some reference to indigenous criminology and the work of Cunneen and Tauri in this area. This would also set the paper up for discussing the use of RJ in these indigenous communities. Again this would need to have a more international focus with reference to indigenous people in the US, Canada, Australia and New Zealand. These revisions / additions are relatively minor so I do hope that the authors are not disheartened by them.
Response: Thank you for the suggestion. After review of their work on Indigenous criminology, we have added a citation of Cunneen & Tauri (2017) to our background section. We feel that this reference fits appropriately in section 1.2. Incarceration and Indigenous Overrepresentation in the United States. As this systematic review serves more as an overview, we will take these suggestions in future work that explores Indigenous RJ models more in-depth.
Round 2
Reviewer 2 Report
Comments and Suggestions for Authors
Thank you for your responses and for taking the suggestions on board - I do hope that you feel that they make this a stronger paper. Having reviewed the revised paper I certainly feel it carries more weight than the original. The authors should be commended for taking both reviewers feedback on board with such gusto.